# Organic carbon inventories in natural and restored Ecuadorian mangrove forests

Amanda G. DelVecchia[1,6], John F. Bruno[2], Larry Benninger[3], Marc Alperin[4], Ovik Banerjee[1,7] and Juan de Dios Morales[5]

[1] Institute for the Environment, The University of North Carolina at Chapel Hill, Chapel Hill, North Carolina, USA

[2] Department of Biology, The University of North Carolina at Chapel Hill, Chapel Hill, North Carolina, USA

[3] Department of Geology, The University of North Carolina at Chapel Hill, Chapel Hill, North Carolina, USA

[4] Department of Marine Science, The University of North Carolina at Chapel Hill, Chapel Hill, North Carolina, USA

[5] Colegio de Ciencias biologicas y ambientales, Universidad San Francisco de Quito, Cumbayá, Diego de Robles y Vía Interoceánica, Quito, Ecuador

[6] Current affiliation: Flathead Lake Biological Station, University of Montana, Bio Station Lane, Polson, MT, USA

[7] Deceased

Corresponding author
Amanda G. DelVecchia,
amanda.delvecchia@umontana.edu

## ABSTRACT

Mangroves can capture and store organic carbon and their protection and therefore their restoration is a component of climate change mitigation. However, there are few empirical measurements of long-term carbon storage in mangroves or of how storage varies across environmental gradients. The context dependency of this process combined with geographically limited field sampling has made it difficult to generalize regional and global rates of mangrove carbon sequestration. This has in turn hampered the inclusion of sequestration by mangroves in carbon cycle models and in carbon offset markets. The purpose of this study was to estimate the relative carbon capture and storage potential in natural and restored mangrove forests. We measured depth profiles of soil organic carbon content in 72 cores collected from six sites (three natural, two restored, and one afforested) surrounding Muisne, Ecuador. Samples up to 1 m deep were analyzed for organic matter content using loss-on-ignition and values were converted to organic carbon content using an accepted ratio of 1.72 (g/g). Results suggest that average soil carbon storage is $0.055 \pm 0.002$ g cm$^{-3}$ ($11.3 \pm 0.8\%$ carbon content by dry mass, mean $\pm 1$ SE) up to 1 m deep in natural sites, and $0.058 \pm 0.002$ g cm$^{-3}$ ($8.0 \pm 0.3\%$) in restored sites. These estimates are concordant with published global averages. Evidence of equivalent carbon stocks in restored and afforested mangrove patches emphasizes the carbon sink potential for reestablished mangrove systems. We found no relationship between sediment carbon storage and aboveground biomass, forest structure, or within-patch location. Our results demonstrate the long-term carbon storage potential of natural mangroves, high effectiveness of mangrove restoration and afforestation, a lack of predictability in carbon storage strictly based on aboveground parameters, and the need to establish standardized protocol for quantifying mangrove sediment carbon stocks.

## INTRODUCTION

The concentration of atmospheric $CO_2$ has increased by forty-percent since the beginning of the industrial revolution and continues to increase concentrations by 2 ppm annually (*Dedysh, Derakshani & Liesack, 2001*; *Le Quéré et al., 2012*). As a result, we face a warming planet, rising seas, changing precipitation patterns, and decreasing biodiversity (2012). Identifying effective, efficient, and politically acceptable approaches to reduce the atmospheric concentration of $CO_2$ is thus one of society's most pressing goals. Reducing atmospheric $CO_2$ via carbon sequestration—transferring carbon to a safe biological or geological reservoir—is one such solution.

Terrestrial vegetation plays a key role in the global carbon cycle as both a sink and a source of anthropogenic $CO_2$: total forest carbon uptake is $2.3 \pm 0.4$ Pg C yr$^{-1}$ (*Pan et al., 2011*), whereas the loss of vegetation via land use change adds $1.1 \pm 0.7$ Pg C yr$^{-1}$. While terrestrial forests as a whole are a net sink, tropical land use change emits $1.3 \pm 0.7$ Pg C yr$^{-1}$ (*Pan et al., 2011*). Conservation of existing vegetation is therefore critical for preventing further carbon emissions as well as for preserving carbon sequestration potential.

Despite the greater area of terrestrial carbon sinks (*Schlesinger, 1997*), coastal carbon sinks have comparable global carbon sequestration values: total global carbon uptake in mangroves, salt marshes, and seagrass beds is estimated at 84–233 Tg C yr$^{-1}$ and uptake in terrestrial systems is estimated at 180.8 Tg C yr$^{-1}$ (*Donato et al., 2011*; *McLeod et al., 2011*). In coastal ecosystems, high rates of uptake reflect high sediment accumulation rates ranging from 18 to 1713 g C m$^{-2}$ yr$^{-1}$ (*McLeod et al., 2011*); organic carbon burial occurs as sediment is accreted vertically during periods such as the present, when sea level is rising (*Ellison, 2008*).

On an aerial basis, mangroves display some of the highest rates of carbon burial and storage among vegetated habitats, sequestering $2.26 \pm 0.39$ Mg C ha$^{-1}$ yr$^{-1}$ and storing an estimated 1,023 Mg C ha$^{-1}$ in aboveground and sediment stores combined (*Donato et al., 2011*; *McLeod et al., 2011*). Their elaborate root structures slow the rate of water movement and thereby create an environment conducive for the settling of clay and silt particles (*Wolanski, 1995*; *Young & Harvey, 1996*). The carbon buried in these systems has been traced to not only autochthonous sources such as litterfall, benthic macroalgae, and root decay, but also imported sources such as seagrass and phytoplankton detritus, showing that mangrove forests provide broad-scale sink benefits (*Kristensen, 2007*; *McLeod et al., 2011*).

International carbon marketing systems such as REDD+ (Reduced Emissions from Deforestation and Forest Degradation) place forest conservation projects in the context of the global carbon offsets market. Such marketing requires accounting for the dynamic nature of accumulation rates over temporal and geographic scales which are still not fully understood (*Alongi, 2011*). However, it is becoming increasingly clear that mangrove

conservation (in a carbon trading context) is more valuable for preventing carbon release from deforestation than for continuously accounting for new sequestration (*Alongi, 2011*; *Donato et al., 2012*; *Fourqurean et al., 2012*).

Mangrove deforestation generates emissions of 0.02–0.12 Pg annually C $yr^{-1}$, the equivalent of 2–10% of emissions from tropical deforestation despite the fact that global mangrove area is <1% of that of tropical forest area (*van der Werf et al., 2009*; *Giri et al., 2011*; *Donato et al., 2012*; *Le Quéré et al., 2012*). Nearly half of the world's mangroves forests have already been cleared, and the recent deforestation rate is roughly 1–3% annually (*Alongi, 2002*; *Bouillon et al., 2008*; *Donato et al., 2011*). Mangroves are usually cleared for development or conversion to aquaculture (*Alongi, 2002*). Upon clearing, both the aboveground biomass and sediment carbon stores are disturbed and/or aerated, increasing microbial activity (*Granek & Ruttenberg, 2008*; *Couwenberg, Dommain & Joosten, 2010*; *Lovelock, Ruess & Feller, 2011*; *Pendleton et al., 2012*).

Though conservation of these ecosystems could be incentived by recognizing both their continuing sink potential and the adverse effects of deforestation via carbon release, the application of existing information to conservation initiatives is limited by a lack of empirical data. Most carbon storage and sequestration studies are from Florida, China, the Indo-Pacific, Australia, and the Brazilian coastline, despite global distribution of mangroves on coastlines between 0 and 30 degrees latitude (*Fujimoto et al., 1999*; *Cebrian, 2002*; *Chmura et al., 2003*). Mangrove storage and sequestration estimates in South America, especially on its Pacific coast, have been extremely rare thus far.

In addition, methodological discrepancies have led to significantly different results which are difficult to interpret. Carbon storage and sequestration quantification is limited by a lack of concurrent data on depth, bulk density, carbon concentration, and sediment accumulation rates (*Alongi, 2011*; *Donato et al., 2011*). Finally, though the value of mangrove conservation can be inferred from previous observations of their natural state, little work has addressed the effectiveness of restoring these ecosystems in terms of carbon storage and sequestration (*Laffoley & Grimsditch, 2009*).

The purpose of the study was to understanding how mangrove carbon storage varies with environmental context. Primarily, we asked how soil carbon standing stocks vary based on forest structure, locations within mangrove patches (defined here as continuous stands of mangroves), and patch land use history. Additionally, we examined how carbon concentration varies with soil depth in a given location and how these concentrations may be most accurately determined. We used these estimates to analyze the carbon storage efficiency of restoring mangroves in sites previously cleared for shrimp farming, and introducing mangroves to replace native vegetation.

## METHODS

### Study sites

We surveyed the forest and collected soil cores at six sites in coastal Ecuador. The sites are located between 0°32′N and 0°38′N and surround the island of Muisne in the Esmeraldas province of Ecuador. This area is unique for its community-driven focus on conservation

and successful restoration of mangrove forests. We selected three natural sites, two restored sites, and one afforested site that have similar geography and comparable patch sizes. All sites are mainly monocultures of red mangroves (*Rhizophora mangle*) with scattered white (*Laguncularia racemosa*) and black mangroves (*Avicennia germinans*) at the fringes.

We determined site histories using a combination of unpublished maps and land use documents from the Jatun Sacha Foundation (a local non-profit conservation organization), interviews with local residents and property owners, and official maps from Ecuador's Instituto Geografico Militar (Andres Leith, pers. comm). The natural sites (Nat A, Nat B, and Nat C) are located in mangrove forest that has been undisturbed for at least three decades (and likely much longer). The restored sites (Rest A and Rest B) were predominantly mangrove forest until the 1980's, at which point they were dredged, diked, and filled for use as shrimp farms until the time of restoration in 2003 (Rest A) or 2000–2002 (Rest B). These sites, having been re-established by planting of red mangrove propagules gathered from existing populations, are characterized by smaller trees with more uniform ages. The afforested site (Aff) is an area that was converted from halophytic ferns to mangrove in 1993.

At each of these sites, we established six plots using a random selection of coordinates. For each plot, we took forestry surveys and outlined a 1 × 1 m quadrat that could be used to take replicate soil core samples. We then mapped site coordinates and used Google Earth to measure the shortest straight-line distance to the mangrove patch edge to determine a rough estimate of distance to the estuarine shoreline. We estimate a measurement error of approximately 30 m on coordinate and distance measurements due to: (a) the difficulty of obtaining satellite signals from within the dense mangrove canopy and (b) inaccuracy in the simple straight line measurements made using Google Earth.

## Forestry surveys

In order to test whether sediment carbon storage varied with forest composition and density, we collected and analyzed forestry data using the protocol outlined in the GOFC-GOLD sourcebook published by REDD (Reduced Emissions from Deforestation and Forest Degradation) (*Pearson, Walker & Brown, 2005*). At each plot, we first delineated a 2 × 2 m quadrat, in which we used a diameter tape to measure the DBH, or 'diameter at breast height' (height 1.3 m) of each tree. In cases where the prop roots typical of *R. mangle* extended above breast height, we took the diameter at 30 cm above the uppermost root connection to the main trunks (*Komiyama, Poungparn & Kato, 2005*). Multiple trunks were individually measured for use in allometric equations but noted as the same tree in tree density calculations (*Clough, Dixon & Dalhaus, 1997*). As per calculations recommended by the GOFC-GOLD Sourcebook, we used a nested plot design to measure total forest biomass. If any DBH exceeded 5 cm, the 2 × 2 m was then extended to a a 7 × 7 m plot, in which we followed the same process for all trees with at least one trunk >5 cm in DBH. Finally, if any trees in this plot exceeded 25 cm in diameter, we extended the quadrat to 25 × 25 m and measured all trees with diameters >25 cm.

We used DBH data to derive aboveground biomass estimates using the species-specific allometric equations recommended by *Komiyama, Ong & Poungparn (2008)*
*Rhizophora mangle*

$$W_{\text{top}} = 0.178(\text{DBH})^{2.47}, \quad r^2 = 0.98, n = 17 \tag{1}$$

*Avicennia germinans*

$$W_{\text{top}} = 0.140(\text{DBH})^{2.54}, \quad r^2 = 0.99, n = 21 \tag{2}$$

*Laguncularia racemosa*

$$W_{\text{top}} = 0.209(\text{DBH})^{2.24}, \quad r^2 = 0.99, n = 17. \tag{3}$$

These equations accounted for all aboveground biomass. Estimates of biomass density at the hectare scale were calculated by scaling up the $2 \times 2$, $7 \times 7$, and $25 \times 25$ m quadrat biomass and tree totals to hectare totals.

## Soil core collection

We collected two soil cores from each of the 36 plots using a 6.69 cm inner-diameter $\times$ 1 m length stainless steel core tube with a sharpened edge. The tube was equipped with a rubber piston held by rope at the top of the soil surface (or water surface if the soil was submerged) to minimize compaction as the core tube was pushed down (Fig. 1). The piston was maintained in place relative to the tube as the core was retrieved from the soil, ensuring soil retention. When the tube reached the soil surface, a rubber plug was inserted into the bottom of the tube.

The core was sectioned in the field by propping the tube on a wooden dowel, removing the upper rubber piston, and pushing down on the core barrel to extrude the soil upward. We sampled 1 cm sections at 6 cm resolution, discarding the uppermost 5 cm as litter fall. Samples were removed using a stainless steel knife run along the top edge of the core tube. We discarded a 5 mm rind from each section to remove soil that may have been mixed due to friction along the wall of the core tube. Soil samples were double sealed in Whirl-paks® and frozen.

## Soil analyses

To obtain bulk soil density, we removed visible root material; decaying plant matter and dead wood were left in the sample. Root removal is necessary to measure soil density, but results in underestimating organic carbon inventories, as the woody root matter accounts for an average of 8.1% of the sample volume in samples where roots was removed. After root removal, we transferred soil samples to tared aluminum foil boats, dried them at 105 °C for 12 h, and reweighed each sample. The drying time was validated by drying a subset of 67 samples for 12, 24, and 48 h; relative differences in mass between 12 h and each of the longer durations were 0.2% and 0.3%, respectively, so 12 h was chosen as an acceptable drying time. Bulk soil density was calculated as the mass of dry soil per volume of bulk soil. Bulk soil volume was calculated for each sample using a 5.59 cm diameter

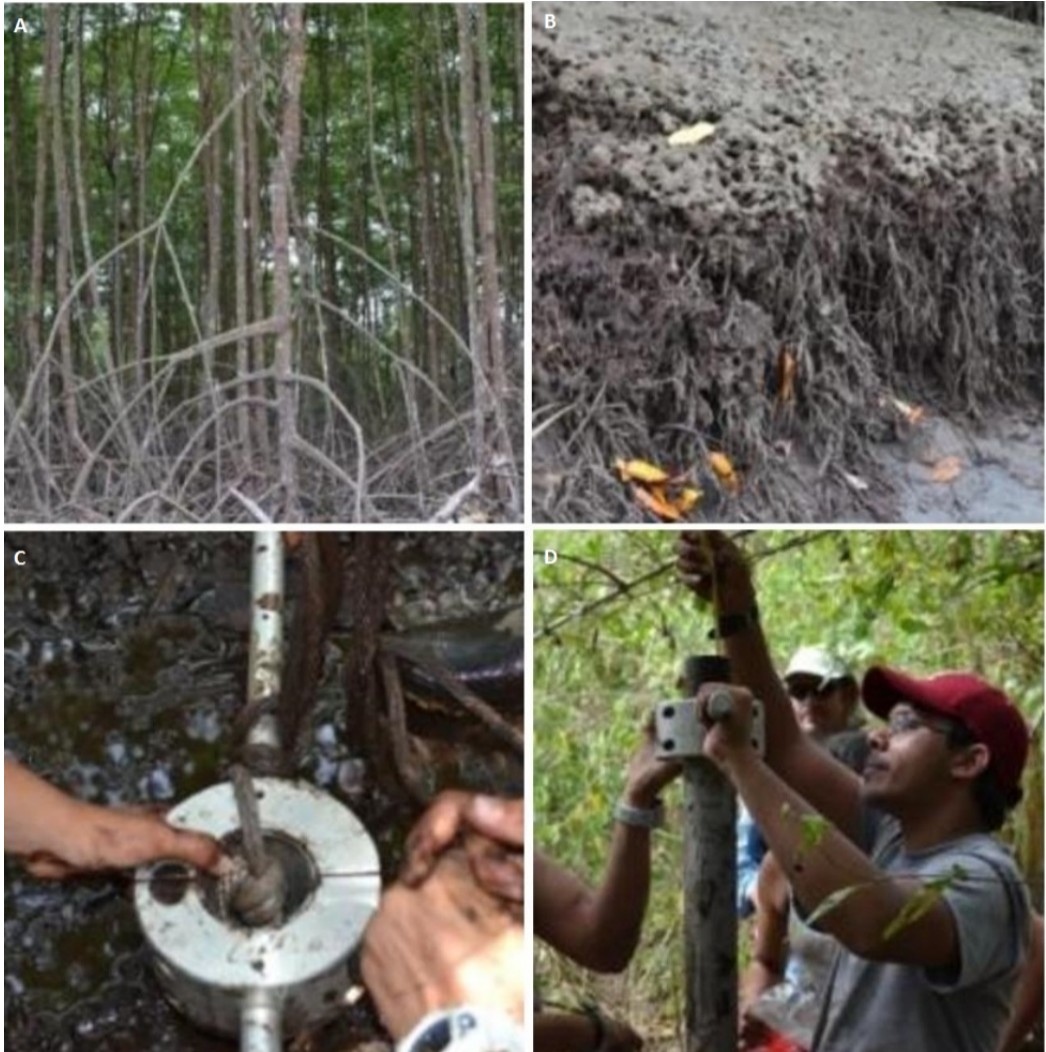

**Figure 1 Sample collection methods.** (A) Prop root structure on red mangroves at Site Nat 2; (B) exposed root structures at low tide show accumulation of sediment; (C) field extrusion method: discard of 5 cm; (D) core tube fully submerged in sediment with rubber piston held at the surface of the ground.

(after discarding rind) and measured core segment height, with the volume of the removed root matter subtracted (root matter volume was measured to a precision of 0.2 cm$^3$ using water displacement).

One replicate core from each plot was processed exclusively for loss-on-ignition (LOI). Dried samples were ground and homogenized using a mortar and pestle until the material could pass through a 2 mm mesh. We transferred the entire dry sample to a tared crucible to burn at 500 °C for 12 h, as recommended by *Wang, Li & Wang (2011)* for non-marine sediments. The reported precision of the LOI method depends on soil type, but is always $<\pm 15\%$ of the measured value (*Wang, Li & Wang, 2011*). We did not measure LOI reproducibility because the entire sample was combusted.

We tested the traditionally assumption that in general, organic matter (represented by LOI) is comprised of 58% organic carbon, yielding a 1.72 conversion factor (*Allen, 1974*). Despite the likelihood that these assumptions provide only approximations, the 1.72 conversion factor has been used to derive global estimates of mangrove carbon storage and sequestration (*Chmura et al., 2003*; *Duarte, Middelburg & Caraco, 2005*; *McLeod et al., 2011*). We examined this assumption by sub-setting samples carbon analysis.

The second replicate core from selected plots—two natural, two restored, and two afforested cores were chosen arbitrarily—was used for total organic carbon (TOC) analysis using a Carlo-Erba Elemental Analyzer. These samples were dried, ground, and homogenized following the procedure described above for LOI. Triplicate 7–10 mg aliquots of each dried and homogenized sample were weighed into tared tin boats and fumed with gaseous HCl to remove inorganic carbon. We followed the method of *Hedges & Stern (1984)* except that we used tin rather than silver sample boats. Tin reacts with HCl vapor to form $SnCl_2$, possibly affecting the tare and causing the boats to become brittle. The reported precision of the TOC method is ±1% of the measured value (*Hedges & Stern, 1984*); however, precision of our TOC analyses averaged ±18% (range: 2–45%), probably related the use of tin boats. This has limited impact on our organic carbon inventories, since they are ultimately related to LOI, which has similar reproducibility. The remaining soil from the second replicate core was analyzed for LOI to provide paired data for forming the TOC-LOI calibration equation.

## Statistical analyses

We averaged sample measurements hierarchically to analyze on the levels of plot, site, and classification. Whenever possible, we averaged measurements by depth from both replicate cores in each plot ($n = 36$), and otherwise used the measurements from a single core ($n = 8$). In cases where we looked at site-specific and classification-specific differences, we first averaged plot measurements to compute averages by depth and/or carbon standing stocks. We took carbon concentration to be the product of individual sample bulk density and %TOC values.

We used the R platform for all statistical analyses. We used the caTools package to compute integrated loess curves (span = 0.5) over the maximum depth interval per set of plot averages to calculate total carbon standing stock per unit area per plot. All integrations began at 5 cm of depth rather than at surface level to avoid the uncertainty introduced by extrapolation, so we may have underestimated the carbon standing stocks. We separately integrated all plots with core measurements >70 cm in depth from 5 to 70 cm to compare sites and classifications without the confounding effect of varying core lengths; we term these integration results *corrected carbon stocks*.

We used linear mixed effects models to assess the effects of site, land history classification, total aboveground biomass, tree density, species composition (percent trees which were red mangroves versus white or black) and distance to estuarine shoreline on the corrected carbon stocks using the NLME package in R. The Akaike information criterion (AIC) was used to determine the best models and parameters using random components

**Table 1 Core sample statistics summarized per site.** Summary statistics per site (means ± standard errors) for a soil depth of 71 cm (corrected carbon stocks) using both the 1.72 conversion factor and our conversion equation (∗).

| Site | Aboveground biomass ($Mg \cdot Ha^{-1}$) | Sediment carbon storage ($Mg \cdot Ha^{-1}$) | Sediment carbon storage (($Mg \cdot Ha^{-1}$)*) |
|---|---|---|---|
| Nat A | 70 ± 18 | 448 ± 143 | 397 ± 175 |
| Nat B | 193 ± 57 | 387 ± 45 | 356 ± 63 |
| Nat C | 39 ± 11 | 386 ± 61 | 374 ± 177 |
| Rest A | 24 ± 5 | 427 ± 54 | 365 ± 97 |
| Rest B | 46 ± 10 | 395 ± 22 | 321 ± 71 |
| Aff | 93.3 ± 1 | 399 ± 22 | 304 ± 67 |

for site and classification values. Additionally, we used a Welch Two Sample *t*-Test to compare natural and restored site standing stocks using the corrected measurements (up to 70 cm and no further).

## RESULTS

### Aboveground biomass (AGB) and tree density

Natural sites and the afforested site had significantly higher AGB and significantly lower tree density than the restored sites (Fig. 3; Table 1). As a mangrove forest matures, the trunks get larger and more dispersed, with larger root boles which overlap in the areas between trees. Similar AGB and tree density in afforested and natural sites suggests that 20 years is sufficient for a mangrove forest in this region to reach maturity; lower AGB and higher tree density at the restored sites suggests that the mangrove forest is still maturing 10 years after restoration. AGB for all sites at the low range of previous estimates across all latitudes (*Alongi, 2002*).

### LOI vs. TOC

LOI values must be converted to TOC to accurately quantify soil organic carbon inventories. The mass lost through combustion includes the non-mineral component of organic matter as well as lattice water in clays and other soil components that are volatile at high temperature; oxygen may be incorporated if nonvolatile oxides form during combustion. LOI is sometimes converted to TOC using the van Bemmelen factor (TOC/LOI = 58%), but numerous studies have shown that the TOC/LOI ratio can range over a factor of two (*Howard & Howard, 1990*) and depends on soil type.

Our results suggested a strong linear correlation between % lost-on-ignition and % total organic carbon content found via carbon analysis (Fig. 2, $R^2 = 0.89$, $p < 0.001$): TOC (%) = 0.87 LOI (%)–5.8. The TOC/LOI ratio (87%) and the y-intercept of LOI vs. TOC suggest that these soils contain almost 7% structural water or minerals that are volatile at 550 C. *Howard & Howard (1990)* found that the TOC vs. LOI linear regression yielded the highest y-intercept (5.64) for gley sediments high in clay content (mean 21.9%). This equation had a linear coefficient of 1.52 and an $R^2$ of 52%.

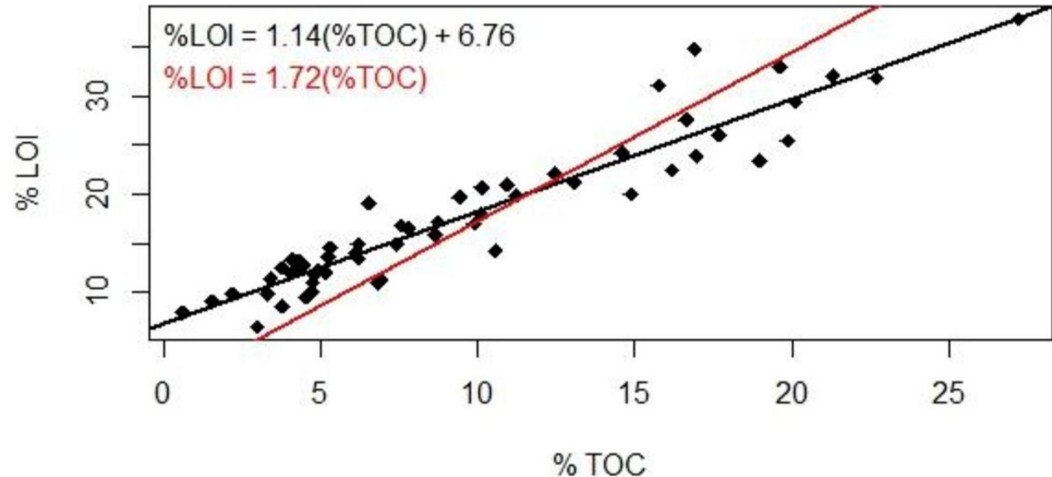

**Figure 2 Loss-on-ignition to percent organic carbon conversion.** Comparison of methods used to derive %TOC values from %LOI measurements. Studies using LOI as a proxy for TOC have traditionally used the 1.72 conversion factor (red); our data suggest the use of a linear regression (black).

The coefficient of variation for our TOC analysis averaged $\pm 18\%$ (range 4%–45%). The reproducibility is well below the analytical precision under optimal conditions ($\pm 2\%$) and may reflect instrumental variability or heterogeneity in the sediment. We therefore present a range of %TOC values calculated using both the van Bemmelen factor and our conversion equation.

## Carbon concentrations

As found in previous studies (*Avnimelech et al., 2001*; *Donato et al., 2011*), our results suggested an inverse relationship between %total organic carbon (%TOC, mass sediment organic carbon per mass sediment) and bulk density (g cm$^{-3}$, mass dry sediment per volume wet sediment):

$$\%\mathrm{TOC} = 6.044(\mathrm{BD})^{-0.775} \text{ (Fig. 4, } n = 442). \tag{4}$$

Overall across all natural and restored sites and depths, median TOC content was found to be 7.38% and average carbon concentration were found to be $55.9 \pm 1.4$ mg OC cm$^{-3}$. There appears to be no consistent change in %TOC in either classification over the 1 m depth interval (Fig. 5). Though a slight decrease is noted in both site classifications from 80–100 cm depth, deeper samples would be necessary to verify whether or not this is a continuous pattern. A significant peak is uniquely present in the %TOC content of the restored sites at approximately 30–60 cm. The same peak holds when measurements are converted to carbon concentrations (Fig. 6). After measurements were converted to corrected carbon standing stocks, results suggested that restored sites might contain more sediment carbon ($411.6 \pm 27.9$ Mg C ha$^{-1}$) than natural sites ($365.3 \pm 23.8$ Mg C ha$^{-1}$), these differences were not statistically significant (two-tailed $t$-test, $p = 0.22$).

Analysis of carbon standing stock using the linear mixed effects models suggested that, as predicted, core length is highly significant ($p < 0.001$) as a predictor of the total

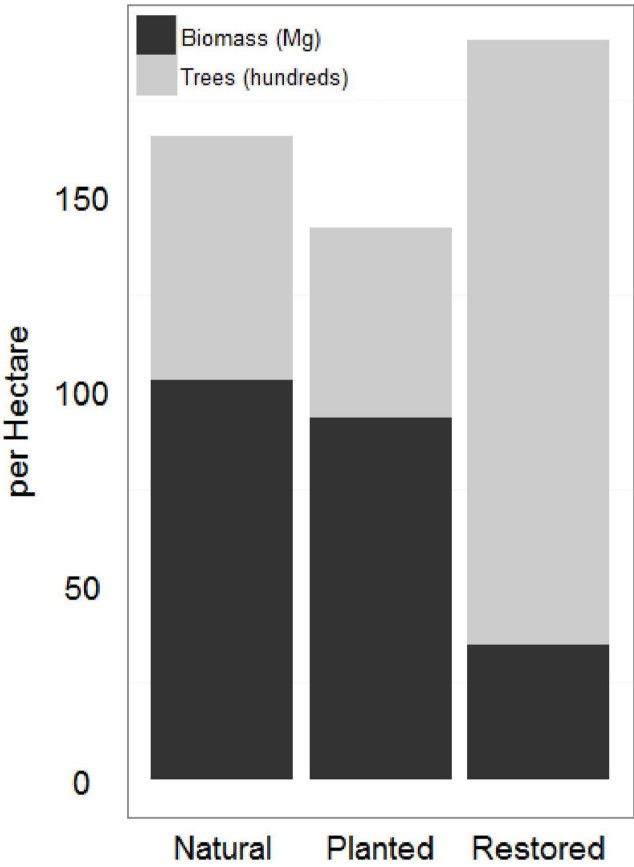

**Figure 3 Restored and natural site biomass and tree density.** Restored sites tended to have higher total trees and lower total biomass estimates than the natural sites. The afforested site overall had fewer trees than either other classification, but more closely resembled the forest structure of the natural sites.

carbon standing stock. No other variable was found to significantly improve the model, e.g., we found no clear evidence of a relationship between distance to estuarine edge, species composition, or site classification and standing stock, even when coordinates were included. Aboveground biomasses and belowground standing stocks by site are presented in Table 1. A Mantel test indicated there was a significant effect of location (UTM coordinates) on total sediment carbon standing stock ($p = 0.05$) but not on total aboveground biomass ($p = 0.89$) but the addition of coordinates to the linear mixed effects models had no effect.

## DISCUSSION

### Carbon standing stocks

Carbon concentration (g C cm$^{-3}$) did not vary significantly between natural or restored mangroves, suggesting that carbon standing stock in ten year old restored mangroves with significantly less aboveground growth is approximately equivalent to stock in natural mangroves that are likely at least 40–50 years old (*Alongi, 2002*). Additionally, carbon concentration did not vary with depth between 5 cm and up to 1 m, suggesting that

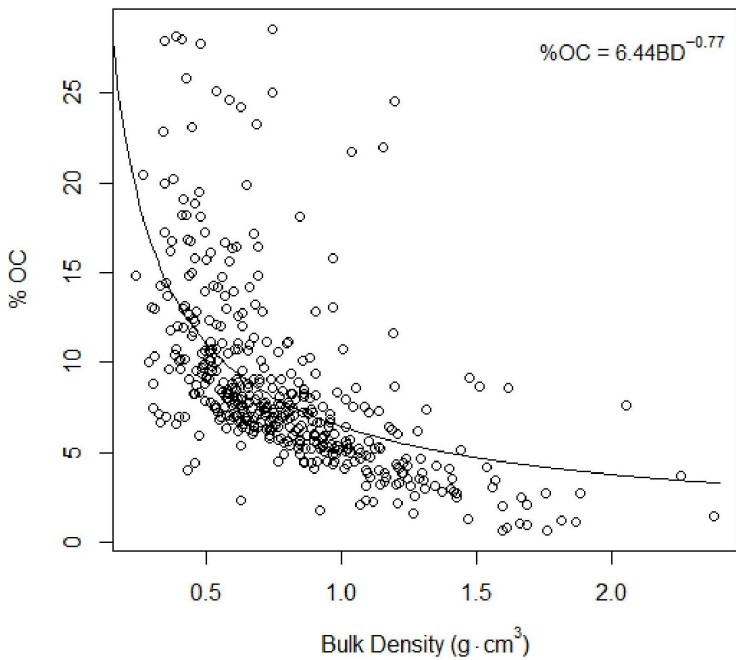

**Figure 4 Total organic carbon vs. bulk density.** Inverse relationship between %TOC and bulk density Eq. (2).

significant decomposition does not occur from the time that the organic carbon is buried until at least the time that this depth represents. This finding was concordant with that of *Donato et al. (2011)*, who concluded that changes in each of these parameters occur deeper than approximately 1 m in depth. Based on published averages of mangrove sediment accretion rates (*Alongi, 2012*; *Breithaupt et al., 2012*) our measured top 90 cm of sediment likely represent $280 \pm 80$ years (95% C.I.) of sediment/carbon accumulation.

We also found that carbon standing stock up to 1 m in depth is strongly correlated with core length. Though this relationship would be expected under the simple assumption that more sediment analyzed implies more carbon to be found, our identification of the relationship emphasizes two points regarding future quantification of globally distributed mangrove stocks. The first is that shallower cores may be useful in calculating carbon stocks up to 1 m in depth, potentially validating extrapolation (to a limit) of studies such as those presented by *Chmura et al. (2003)* which included measurements from up to 0.5 m depth. The second point is that calculation of carbon stocks requires an understanding of soil depth as it varies in mangroves globally, as slight changes in this depth measurement have strong implications for the calculation of total sediment carbon stock. Studies which address overall sediment depth have to this point been rare, as can be seen in several meta-analyses and recommendations for future research (*Chmura et al., 2003*; *Laffoley & Grimsditch, 2009*; *Donato et al., 2012*).

Additionally, we found that neither forest structure and composition nor distance from the seaward edge are significant predictors of carbon standing stock up to 70 cm in depth. Though the clear differences in aboveground biomass with forest maturity likely

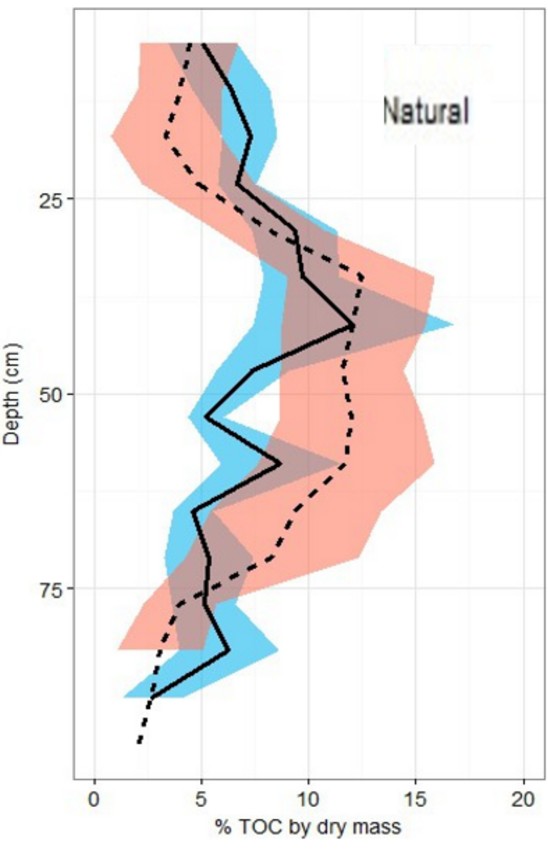

**Figure 5  Total organic carbon profiles of natural and restored sites.** %TOC profiles (means ± standard error using 1.72 conversion factor) suggest negligible differences between the two main site classifications, but a slight peak in the restored site profile is noted between 40 and 60 cm of depth.

influence immediate accretion and litterfall rates, the dynamic nature of these forests (i.e., tendency for scour and/or deposition during storm surges) would prevent us from detecting those effects in measurements of carbon storage over long time periods without intensive sampling.

## Natural vs. restored and afforested site parameters

We found that natural, restored, and afforested mangrove sites are equally important in terms of current carbon standing stock, emphasizing the value of preservation of relatively young forests as well as old growth stands–disturbance of either classification would aerate similar quantities of sediment organic carbon. We found a 12.6% higher mean value for restored sites than natural sites (365 Mg C ha$^{-1}$), despite the clearly later successional stages of the natural sites. This implied that mangrove restoration of shrimp farms is effective at restoring ecosystem function, at least in terms of carbon sink potential seen in the decade post-restoration. This finding contradicts that of *Osland et al. (2012)*, who found lower rates of carbon storage in restored mangroves.

The higher mean carbon stock of restored sites appears to be due to a peak in concentrations at approximately 30–60 cm deep. Because the same peak was not present

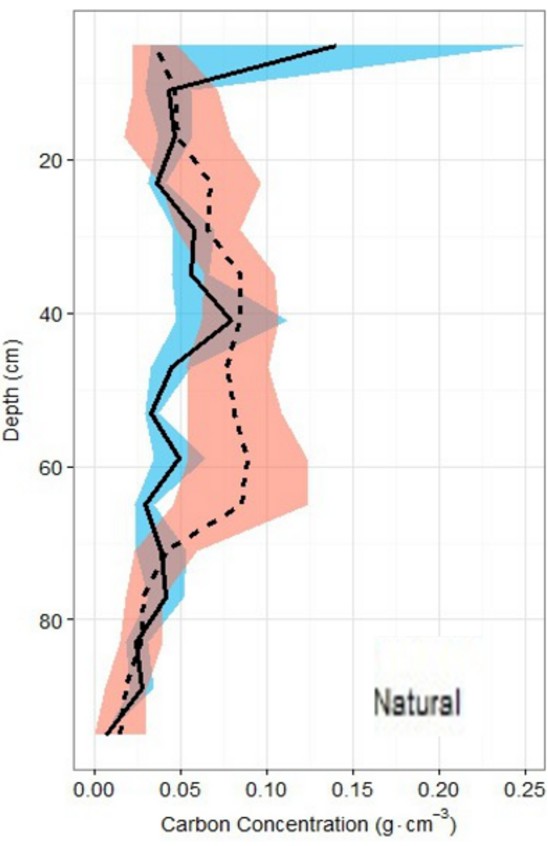

**Figure 6 Carbon density profiles in natural and restored sites.** Carbon concentration profiles (means ± standard error using 1.72 conversion factor) suggest negligible differences between the two main site classifications, but again, a slight peak in the restored site profile is noted between 20 and 60 cm of depth.

in the afforested site profile, it could represent a relic of the shrimp farm history of the restored sites. Higher concentration values could result from shrimp carcasses, 'fertilizer', or feces that accumulated during the farming period and were buried when the mangroves were replanted and began to accrete sediment. Because the shrimp farms are excavated and form low spots in the landscape, they are prime areas for sedimentation as currents flow in and pool, especially if mangroves are present to encourage the trapping of sediment particles. If this rapid sedimentation indeed occurred, the theoretical 3 cm yr$^{-1}$ of accretion that our results would suggest would rapidly place the high quantities of organic matter in an anaerobic environment and potentially reduce the rate of decomposition. If this is indeed the case, restoration of shrimp farm plots to mangroves mitigates much of the change that original shrimp farm construction might have caused.

## Current context and future concerns

Geographically, our study is unique in that it provides the first estimate of mangrove carbon storage on the Pacific coast of South America. Putting this into a global context, our results are concordant with those published in meta-analyses that synthesize studies mainly from Southeast Asia and Florida (Table 2). Our measurements come from equatorial

**Table 2 Comparison of carbon storage estimates to global statistics.** Comparison of natural and restored site carbon storage estimates to previously published estimates.

| Source | Region(s) | Mean soil carbon concentration (g C cm$^{-3}$) | Core length |
|---|---|---|---|
| *Chmura et al. (2003)* | Global | 0.055 | 0.5 m |
| *Donato et al. (2011)* | Indo-Pacific | 0.038 (Estuarine), 0.061 (Oceanic) | Variable, up to 3 m |
| *Pendleton et al. (2012)* | Global | 0.015–0.115 | Variable |
| Natural, this study | Ecuador | 0.055 ± 0.002 | 0.65–1 m |
| Restored, this study | Ecuador | 0.058 ± 0.002 | 0.65–1 m |
| Afforested, this study | Ecuador | 0.056 ± 0.002 | 0.65–1 m |

mangroves, which are thought to be among the highest productivity globally (*Alongi, 2002*). This potential geographical variation should be considered in extrapolations to global mangrove carbon storage.

Mangrove restoration is becoming increasingly attractive as we search for ways to mitigate climate change. Though preservation of existing carbon stocks is a clear way to prevent additional emissions from deforestation, restored shrimp farms display even higher carbon standing stocks than mangroves, which are thought to have some of the highest rates globally. It is likely that having mangroves present promotes burial—rather than disturbance and aeration—of the carbon present. Because there are very few studies of mangrove restoration potential in terms of carbon storage, a proper evaluation will require additional studies in other areas where restoration may be a viable option, perhaps with a specific focus on those regions (unlike ours) where monocultures do not occur naturally as well as in restored zones. Long-term monitoring will be needed to verify the continued storage of the carbon peaks we observed. Our study demonstrates the potential for mangrove restoration to effectively sequester carbon.

## ACKNOWLEDGEMENTS

This work could not have been completed without the tremendous help of Katie Dubois, Andrew Chan, Kaitlyn Ferguson, Spencer Scheidt, Andres Ledergeber, Rachel Gittman, Barbara MacGregor, Jack Stanford, and the staff and volunteers at Congal Biomarine Station. Thanks also go to the Jatun Sacha Organization and Diego Quiroga for help establishing a field base.

John Bruno is the sole owner of The Blue Carbon Project, a start up corporation developed to facilitate mangrove restoration and conservation through the sale of carbon offset credits.

### Funding

Research was funded in part by a Summer Undergraduate Research Fellowship from the Office for Undergraduate Research at the University of North Carolina at Chapel Hill, and a Watts-Hill Award from the Institute for the Environment at the University of North

Carolina at Chapel Hill. The funders had no role in study design, data collection and analysis, decision to publish, or preparation of the manuscript.

### Grant Disclosures

The following grant information was disclosed by the authors:
The Office for Undergraduate Research at the University of North Carolina at Chapel Hill.
Institute for the Environment at the University of North Carolina at Chapel Hill.

### Competing Interests

John Bruno is an Academic Editor for PeerJ and is the sole owner of The Blue Carbon Project, a start up corporation developed to facilitate mangrove restoration and conservation through the sale of carbon offset credits.

### Author Contributions

- Amanda G. DelVecchia conceived and designed the experiments, performed the experiments, analyzed the data, wrote the paper, prepared figures and/or tables, reviewed drafts of the paper.
- John F. Bruno and Marc Alperin conceived and designed the experiments, analyzed the data, wrote the paper, reviewed drafts of the paper.
- Larry Benninger conceived and designed the experiments, analyzed the data, contributed reagents/materials/analysis tools, wrote the paper, reviewed drafts of the paper.
- Ovik Banerjee and Juan de Dios Morales performed the experiments, reviewed drafts of the paper.

### Data Deposition

The following information was supplied regarding the deposition of related data:
  FigShare: Plot Data. Amanda Delvecchia. Retrieved 19:34, Apr 29, 2014 (GMT)
  http://dx.doi.org/10.6084/m9.figshare.1009049—See more at: http://figshare.com/preview/_preview/1009049#sthash.vL3rFcuS.dpuf.

### Supplemental Information

Supplemental information for this article can be found online at http://dx.doi.org/10.7717/peerj.388.

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
