# Peer review of "Organic carbon inventories in natural and restored Ecuadorian mangrove forests"

_PeerJ, doi:10.7717/peerj.388_

## Round 0.1 · original submission · Major Revisions

This paper has been thoughtfully assessed by two reviewers. Both reviewers believe that the study addresses an important topic. I concur. While reviewer # 2 has not identified any major flaws, this is not the case for reviewer # 1. The latter identifies two problems: the omission of data on visible root material, and the lack of data (for both LOI and OC) from replicate samples within the same soil sample. While I am less concerned about the latter, the former does worry me. The paper reports on data that may well be used as justification for attracting finance for restoration initiatives. Therefore, scientific rigor is of the essence.

If you can convince me in your rebuttal that the concerns expressed by reviewer # 1 are not fatal, I will be happy to support publication of this paper in Peer J.

Reviewer 1 ·

Basic reporting

Introduction
Line 45 - not all mangroves have elaborate root structures and some mangrove deposits are nearly all organic with little clay or silt
L 74 - I am surprised not to see Spaulding’s atlas cited here
L 87 - It is not carbon concentration (%) that we need to know, but C density – authors misuse the term concentration throughout the manuscript
Results
L 223 - trunks become more dispersed with age?? Or do you mean there is loss of trunks over time? And root boles (? What are these) and larger than what?
L 226 - a verb is missing here
L 231 – Once in my clay mineralogy class I calculated the loss of mass attributable to lattice water in clays and it was negligible – I challenge authors to do the same.
L 237 - The relationship reported here is the reverse to what is shown on the graph – dependent and independent variables are switched.
L 253 - Variation of % C tells us little – authors should focus on C density
L 260 - a figure relaying these results would be informative
L 270 - Why would simply UTM coordinates be a predictor? What does it relate to?

Discussion
L283 – there has been no measurement of sediment accumulation rates and this sentence should be deleted
L 286 – are authors actually stating the obvious, that the longer the core the greater the C stock measured??
L 312 - rates have not been measured so this study cannot contradict Osland et al.
L 316 - Was there no stratigraphic evidence of a shrimp pond in the cores?

Keywords – as soil accretion was not measured this would be misleading as a key word

Experimental design

Study area
a map would be useful
30 m sounds like a great deal of error

Methods
Why not use the same quadrats for forest survey and soil sampling?
L 103 - How can a personal communication be a supporting ref foe map and interviews?
L 175 – mangrove sediments are marine samples (they have lots of forams and can have high salinity), thus why use a protocol for non-marine samples?
L 182 – Chmura et al used the calibration of LOI published by Craft et al.
L 190 – If tin is a problem, then why was it used? Would the formation of SnCl2 bias analyses?
L 205 – I do not know what an integrated loess curve is or loss curve for that matter

Validity of the findings

This study could have great value in providing novel data on Blue Carbon stocks on the Pacific coast of South America and how stocks might recover after a mangrove forest is restored. However, there are two major flaws in the methods that prevent the authors from concluding anything about differences in soil carbon stocks.
First, investigators removed “visible root material”, but left dead wood. Since mangrove roots are woody how did they distinguish between live and dead roots. Root material generally is not removed when calculating soil C stocks (see Warren et al. 2012 and papers published previously by those who sampled mangrove soils). One might expect more root material in older sites, thus making recovery of C at restored sites appear to be faster than it actually is, removing support for a major point of the paper that carbon stocks in afforested/restored patches are equivalent. This might be remedied if investigators saved the root material – it could then be analyzed. They seem to have measured its volume. Although dry mass would be better, perhaps C could be estimated based upon reported wood densities for mangroves.
Second, there is no true replication (see line 176). There should be multiple samples for LOI and OC from the same soil sample. It is well known that soils are spatially variable. Thus, it is inappropriate, as authors did, to pair LOI analysis from one core to C analysis of another core.

Additional comments

General Comments
There are references missing and I think the figures are out of order.
Grammar needs considerable cleaning up – with this extensive author list one should be able to do this.
Supporting references are sometimes missing a year and sometimes a year is found dangling without the authors in a sentence, and et al. is not used consistently

Ref cited
Warren et al. 2012. Biogeosciences, 9, 4477–4485, 2012

·

Basic reporting

The basic reporting in my view is logical, clear and overall satisfactory, apart from two points:

1. Minor citation issues, where references are not cited accurately or not at all. These are:
i. Page 2, line 5: referenced as “2012” - I could not find this reference on the internet, please provide better details or leave out.
ii. Page 2, line 8: referenced as “Kauffman et al.” - Authors need to cite full reference details in this and all the other instances where this reference is referred to in the text or leave out.
iii. Page 10, line 226: provide references for the statistical packages. Both base R and the relevant package caTools are not referenced in the manuscript.

2. Minor grammar and language issues that need to be rectified:
a. Page 9, line 198: “traditional” rather than “traditionally”
b. Page 11, line 251-252: I don't understand the last sentence (“AGB for all sites...”) in this paragraph, am I missing a verb?

Experimental design

I can find no fatal flaws in the experimental design of the study. Considering the intricacies involved in sampling and analysis of soil for organic carbon content, I think the authors did a good job. I have however, two broad comments.

Firstly, it is a pity that tin rather than silver tared boats were used for samples undergoing the HCl fume treatment for removal of inorganic carbon before being subjected to TOC analysis. The authors admitted this situation truthfully, and as mentioned in the manuscript, it did not affect the validity of the results, nor the conclusions drawn from these. It may encourage future researchers to keep to the original recommendations, should they use this particular method for similar studies.

Secondly, even though it probably was not a major objective of the study, if the authors sampled some additional degraded sites (or old shrimp-farming sites) in the same way as the other sites in the study, it would have added a valuable dimension to the paper. This could have provided better estimates of total carbon sequestration potential of local mangrove restoration initiatives. Although the potential gain in aboveground carbon through restoration is obvious, it is not so clear how much the belowground carbon pools may gain through such action, especially since shrimp-farming may contribute to belowground carbon pools as the authors suggest in their discussion on the results from the restored sites.

Validity of the findings

From what I can determine, the results of the study are based on rigorous and accepted statistical analysis procedures, and the conslusions drawn from these are sound. In my opinion, this study is important in that it augments currently available studies on the subject and contributes to closing the large gap in scientific knowledge of belowground carbon dynamics in mangrove forests globally. I recommend that the study be published with minor revisions as outlined above.

Additional comments

I find it interesting that the afforested site yielded similar carbon pool results to that of natural and restored forests. Can it be that the halophytic ferns that were present before afforestation may represent some successional stage in natural mangrove forest regeneration ?

---

## Round 0.2 · accepted · Accept

I am happy to accept the revised version of this paper. I can now see the logic of excluding root biomass from the analyses.